# Prospective Evaluation of Local Sustained Release of Celecoxib in Dogs with Low Back Pain

**DOI:** 10.3390/pharmaceutics13081178

**Published:** 2021-07-30

**Authors:** Tijn Wiersema, Anna R. Tellegen, Martijn Beukers, Marijn van Stralen, Erik Wouters, Mandy van de Vooren, Nina Woike, George Mihov, Jens C. Thies, Laura B. Creemers, Marianna A. Tryfonidou, Björn P. Meij

**Affiliations:** 1Department of Clinical Sciences, Faculty of Veterinary Medicine, Utrecht University, Yalelaan 108, 3584 CM Utrecht, The Netherlands; t.wiersema@uu.nl (T.W.); a.r.tellegen@uu.nl (A.R.T.); m.beukers@uu.nl (M.B.); m.vandevooren1@gmail.com (M.v.d.V.); 2Image Sciences Institute, University Medical Centre Utrecht, Heidelberglaan 100, 3584 CX Utrecht, The Netherlands; mstrale2@umcutrecht.nl; 3Anicura Dierenziekenhuis Dordrecht, Jan Valsterweg 26, 3315 LG Dordrecht, The Netherlands; erik.wouters@anicura.nl; 4DSM Biomedical, Koestraat 1, 6167 RA Geleen, The Netherlands; chemie_nina@web.de (N.W.); george.mihov@dsm.com (G.M.); jens.thies@dsm.com (J.C.T.); 5Department of Orthopaedics, University Medical Centre Utrecht, Heidelberglaan 100, 3584 CX Utrecht, The Netherlands; l.b.creemers@umcutrecht.nl

**Keywords:** non-steroidal anti-inflammatory drugs, biomaterials, controlled release, intervertebral disc degeneration, microspheres, degenerative lumbosacral stenosis

## Abstract

Back pain affects millions globally and in 40% of the cases is attributed to intervertebral disc degeneration. Oral analgesics are associated with adverse systemic side-effects and insufficient pain relief. Local drug delivery mitigates systemic effects and accomplishes higher local dosing. Clinical efficacy of intradiscally injected celecoxib (CXB)-loaded polyesteramide microspheres (PEAMs) was studied in a randomized prospective double-blinded placebo controlled veterinary study. Client-owned dog patients suffering from back pain were treated with CXB-loaded (*n* = 20) or unloaded PEAMs (“placebo”) (*n* = 10) and evaluated by clinical examination, gait analysis, owners’ questionnaires, and MRI at 6 and 12 weeks follow-up. At 6 and 12 weeks, CXB-treated dogs experienced significantly less pain interference with their daily life activities compared to placebo. The risk ratio for treatment success was 1.90 (95% C.I. 1.24–2.91, *p* = 0.023) at week 6 and 1.95 (95% C.I. 1.10–3.45, *p* = 0.036) at week 12. The beneficial effects of CXB-PEAMs were more pronounced for the subpopulation of male dogs and those with no Modic changes in MRI at inclusion in the study; disc protrusion did not affect the outcome. It remains to be determined whether intradiscal injection of CXB-PEAMs, in addition to analgesic properties, has the ability to halt the degenerative process in the long term or restore the disc.

## 1. Introduction

Back pain is a common clinical problem affecting millions of people globally [1]. In 40% of the patient cases, chronic low back pain is attributed to intervertebral disc (IVD) degeneration [2,3]. The IVD lies between the vertebrae and its main function is to transmit compressive forces, and provide mobility and stability to the spinal column. The core of the IVD, the nucleus pulposus (NP), is contained by the annulus fibrosus (AF), and the two cartilaginous endplates (EPs) of the adjacent vertebrae [4]. IVD degeneration and discogenic pain [5,6] are associated with the local production and influx of pro-inflammatory cytokines, cells, and nociceptive mediators [7], including prostaglandin E_2_ (PGE_2_) and nerve growth factor (NGF) [8,9]. Inflammation promotes catabolic changes in the NP, leading to loss of proteoglycan and water content, which impairs biomechanical function. In an advanced stage, this can lead to a decrease in disc height and spinal instability [10,11]. Concomitant damage to the EPs can lead to reactive bone marrow lesions, which have been shown to be an important factor in low back pain [12]. These pathological changes in the EPs can be visualized in MRI and are referred to as Modic changes [12].

At present, therapy for chronic low back pain related to IVD degeneration is mostly aimed at reducing clinical symptoms and slowing down disease progression. Oral nonsteroidal anti-inflammatory drugs (NSAIDs) are commonly used for the treatment of discogenic pain. However, due to the avascular nature of the IVD there is limited tissue penetration in the IVD and, consequently, the clinical effect is limited [13,14,15]. Prolonged intake of oral NSAIDs is associated with gastrointestinal and cardiovascular side effects [16,17]. Local drug delivery by intradiscal injection can avoid these systemic side effects and ensures optimal drug exposure of the target tissue, the IVD. Biomaterials can facilitate local drug delivery and extend treatment duration by prolonging the presence of drugs in the affected IVD [18].

Several in vitro and preclinical datasets were obtained with an emerging drug delivery platform based on biodegradable amino acid-based polyesteramide microspheres (PEAMs) [19]. Their degradation occurs predominantly via enzyme-mediated hydrolysis as a result of the specific ester to amide bonds ratio [20,21]. The PEA polymers and the products of their bio-degradation are well tolerated in vivo [22] and release celecoxib for at least 80 days in vitro [23], making them especially suitable for prolonged release. Furthermore, PEAMs have been safely administered to the IVD with small diameter needles, caused no further degeneration after injection in mild degenerated IVDs of experimental dogs [24,25], and slowed degenerative changes in the NP and adjacent subchondral bone of the treated IVDs [26].

Considering the advantages of PEAM-based local delivery of CXB, it remains to be determined whether this therapeutic approach would also be effective and provide long term analgesia in a clinical setting prior to bringing this concept to first-in-man studies for patients suffering from discogenic pain. Intradiscal application of a PCLA-PEG-PCLA platform hydrogel-based CXB delivery system demonstrated beneficial clinical effects in a proof-of-concept veterinary study employing dogs that suffered from low back pain [27] due to degenerative lumbosacral stenosis (DLSS) [28]. For the purpose of clinical translation, the dog can serve as a representative model for translational studies to humans. IVD degeneration and disc-related low back pain in the dog show many similarities to human IVD degeneration at the level of clinical presentation, macroscopic and microscopic appearance, diagnostics, and treatment [29,30]. Within this context, COX-2 inhibitors are also broadly used in the pain management of dog patients suffering from low back pain [28,31], and long-term use of oral analgesics is also associated with various side effects and drug-drug incompatibility [32].

The main aim of the present study was to determine the clinical efficacy of local delivery of CXB-loaded PEAMs compared to unloaded PEAMs (“placebo”) in client-owned dog patients suffering from low back pain due to DLSS. For this purpose, a randomized prospective double-blinded placebo controlled veterinary clinical study evaluated the clinical efficacy and adverse effects of intradiscal injection with CXB-PEAMs by clinical examination, gait analysis using fore plate, owners’ questionnaires, and MRI.

## 2. Materials and Methods

### 2.1. Preparation of PEA Microparticles

PEA was synthesized according to a previously published method [33]. Briefly, the PEA polymer was prepared via solution polycondensation of di-p-toluenesulfonic acid salts of bis-(α-amino acid) α,ω- diol diesters, lysine benzyl ester, and di-N-hydroxysuccinimide sebacate in anhydrous DMSO (Sigma-Aldrich, Saint Louis, MO, USA). The polymer was isolated from the reaction mixture in two precipitation steps and characterized by means of proton NMR and size exclusion chromatography [26,34]. To generate CXB-loaded PEAMs, the drug was added to 5 wt% solution of PEA in dichlormethane (Merck Millipore, Burlington, MA, USA). After homogenization, the solution was sonicated in a water bath for 3 min. The PEA-CXB solution was then emulsified in 20 mL of water phase (1 wt% polyvinyl alcohol, 2.5 wt% NaCl; Sigma-Aldrich, Saint Louis, MO, USA) by the use of an ultraturrax (IKA, Staufen, Germany), stirring at 4000 rpm (8000 rpm for empty PEAMs) for 3 min. After emulsification, particles were hardened overnight under air flow. Before washing, particles were cooled with an ice bath for 1 h and washed with solution of Tween 80 (Merck, Darmstadt, Germany). Excess of surfactant was removed by centrifugation. Before freeze-drying, particles were suspended in Tween 80 solution to reach the right concentration of particles per volume. Once dried, the PEAMs were weighed in single high-performance liquid chromatography vials to the approximate amount of 70 mg PEAMs and γ-sterilized on dry ice [26]. CXB loading was 20% resulting in 14 mg CXB per vial. Single vials were prepared so that directly prior to intradiscal injection, PEAMs in a single vial were freshly re-suspended in 3 mL 0.9% sterile saline for intradiscal application.

### 2.2. Study Design

This study was conducted with the approval of the Ethical Committee of the Department of Clinical Sciences, Faculty of Veterinary Medicine of Utrecht University (trial number AVR 18–10, approval date 29 December 2017). It was conducted as a prospective, randomized, double-blinded, placebo-controlled clinical trial. All dogs were client-owned and written consent of the owner was obtained before study enrolment. A random number table was created for 30 dogs (Excel 2016, Microsoft, Redmond, WA, USA). A power analysis was performed on historical data from our institution a priori, to calculate the minimal required group size with an α of 0.05, power of 0.9, and effect size of 1.32. Twenty dogs were allocated to the treatment group (CXB-PEAM) and ten dogs were allocated to the control group (23.3 mg/mL PEAMs). Dogs were divided into weight groups to determine the injection volume: <30 kg dogs received 75 µL (350 µg CXB); between 30–45 kg, 100 µL (466 µg CXB); >45 kg, 125 µL (583 µg CXB). The owners and treating veterinarians were unaware of group allocation. Seven days prior to the inclusion in the study and baseline measurements, the administration of all pain medications was discontinued. Follow-up visits were planned at 6 and 12 weeks after intradiscal injection. Open label follow-up of the primary read out parameter (the questionnaire sumscore) was conducted at 24 weeks to assess clinical outcome according to the owner’s perception.

### 2.3. Inclusion Criteria

Client-owned dogs with DLSS that were presented at the academic veterinary hospital of Utrecht University were included. Inclusion criteria were: dog patients with history of low back pain for at least 6 weeks that was refractory to oral pain medication for >4 weeks, or if oral pain medication caused side effects. Diagnosis of DLSS was confirmed using magnetic resonance imaging (MRI) and IVD degeneration at L7-S1 was graded as Pfirrmann grade II-IV on T2-weighted images. Exclusion criteria were: previously performed surgery on IVD of interest, active discospondylitis, lumbosacral fracture, or spinal neoplasia. Prior to admission of the study, all dogs underwent a full clinical examination by a board-certified orthopedic surgeon (B.P.M.), consisting of a general physical, orthopedic, and neurologic examination. Clinical examination ruled out other causes (e.g., hind limb joint diseases) or locations (e.g., abdominal or pelvic pain) as origin of pain that may resemble low back pain.

### 2.4. Magnetic Resonance Imaging

MRI scans were obtained using a Philips Ingenia 1.5T MRI (Philips, Eindhoven, The Netherlands). The entire procedure was carried out under general anesthesia. Premedication consisted of intravenous (I.V.) administration of butorphanol (0.2 mg/kg) (MSD, Kenilworth, NJ, USA) and dexmedetomidine (5 μg/kg) (Vetoquinol, Magny-Vernois, France), followed by induction of anesthesia with I.V. administration of propofol (1 mg/kg) (Fresenius Kabi, Huis Ter Heide, The Netherlands). All dogs received an endotracheal tube, and maintenance of anesthesia was achieved by administration of isoflurane (2% *v*/*v*) (Zoetis, Capella aan den Ijssel, The Netherlands), delivered in a 1:1 oxygen:air mixture. The dogs were positioned in dorsal recumbency with the pelvic limbs extending caudally. The MR protocol included a sagittal and transverse T2-weighted Turbo Spin Echo (time of repetition (TR) = 2500, time of echo (TE) = 110 ms, slice thickness = 2.5 mm) sequence, a T1-weighted Turbo Spin Echo (TR = 400 ms, TE = 8 ms, slice thickness = 2.5 mm) sequence, and a sagittal multiple spin-echo T2w sequence for quantitative T2 mapping (M.vR) (using custom script in MeVisLab v3.1, MeVis Medical Solutions AG, Bremen, Germany) with a field of view (FOV) = 75 × 219 mm, acquisition matrix  =  96 × 273, slice thickness  =  3 mm, TR  =  2000. Eight echoes were acquired with TE = 13 to 104 ms with 13 ms echo spacing. All images were assessed by a board-certified veterinary radiologist (M.B.) (Enterprise Imaging, version 8.1.2, Mortsel, Belgium). The L6-L7 and L7-S1 discs were graded according to the Pfirrmann grading system [35] on T2 weighted images obtained prior to injection (all dogs) and at 12 weeks (the L6-7 and L7-S1 discs of dogs in the CXB-PEAM group). Mean T2 relaxation times were determined (M.vV) in an oval region of interest (ROI) in the lumbosacral IVDs on midsagittal T2-mapping images. The disc height index (DHI) was measured (T.W.) on the MR images prior to injection in all dogs and at 12 weeks in the CXB-PEAMs-treated dogs using the method described by [36]. At the 3 month follow-up time point, when the labels were opened, the dogs treated with CXB-PEAMs underwent a second MRI, whereas not all owners of the placebo group were eager to submit their dogs to an additional anesthesia for MRI, knowing that their dog had received the placebo.

The descriptive MRI findings commonly reported in DLSS [28] were recorded by the veterinary radiologist (M.B.) blinded to the treatment and time point:(a)Disc protrusion at L7-S1 graded as <25% or ≥25% stenosis of the spinal canal.(b)Pfirrmann grading [36].(c)Presence of Modic changes according to Modic et al. (1988) [37,38].(d)Spinal nerve swelling.(e)Intervertebral foraminal stenosis.(f)Ventral or lateral spondylosis deformans at the lumbosacral junction.(g)Degenerative joint disease of the facet joint.

### 2.5. Intradiscal Injection

The intradiscal injection was performed under CT guidance (B.P.M., A.R.T., T.W.) with the same anesthesia protocol as for MRI. The dog was placed in sternal recumbency with hyperflexion of the lumbosacral region. The lumbosacral site was clipped and prepared aseptically. The lumbosacral space was located through palpation of the iliac crests, the tuber coxae, and the spinous process of L7 and sacrum. A 20G epidural needle (4509757-13, Braun, Melsungen, Germany) was inserted through the skin until bone or ligamentum flavum was encountered. The depth and placement of the needle was checked by CT (Siemens Somatom Definition AS, Siemens Healthcare, Erlangen, Germany) and adjusted when necessary. The needle was then advanced through the ligamentum flavum until the dorsal AF was reached. When correct placement was again confirmed by CT, the stylet was removed and a 12 cm long, 27 G needle (7803-01, Hamilton, Bonaduz, Switzerland) was inserted in the epidural needle that acted as a guide to position the tip of the 27 G needle on top of the AF. The 27 G needle was subsequently advanced through the dorsal AF into the center of the NP and the correct depth of the 27 G needle was confirmed with CT. Finally, a 100 μL gastight syringe (7656-01, Hamilton, Bonaduz, Switzerland) was connected to the 27 G needle and the PEAM suspension was slowly injected. After the syringe was emptied, and prior to removal, CT was repeated to verify the same position of the needle tip in the center of the NP. The needle was then slowly retracted through the AF, allowing for the collagen fibers to close behind the needle to prevent leakage of the PEAMs outside the NP compartment. Physical activity was limited to short leash walks in the first week after treatment. Thereafter, owners could gradually increase activity as before the start of the study, if the dog tolerated this.

### 2.6. Kinetic Gait Analysis

Prior to the intradiscal injection, and at 6, 12, and 24 weeks after the intradiscal injection, ground reaction forces (GRF) were measured (T.W., A.R.T.) with a quartz crystal piezoelectric force plate (Kistler type 9261, Charnwood Dynamics Limited, Rothley, UK) together with the Kistler 9865E charge amplifiers, as described previously [39,40]. Measurements were obtained with a frequency of 100 Hz. GRFs were measured in the mediolateral (Fx), craniocaudal (Fy), and vertical (Fz) direction. The Fz was calibrated with a standard weight before each recording session. The velocity with which the dog walked on the runway was measured using two photoelectric detectors. Measurements in which both a thoracic limb and the ipsilateral pelvic limb had contact with the plate were included. A minimum of ten recordings per side were used for data processing. All forces were normalized for the body weight of the dog. Low back pain in dogs does not primarily present as unilateral lameness but may lead to a shift in forces from the hind- to the frontlimbs. Therefore, ratios between pelvic (P) and thoracic (T) limbs were calculated: P/T Fy−, P/T Fy+, and P/T Fz+ as a measure of hind limb function [39,41].

### 2.7. Owner Assessment of Pain Related Behaviour

Questionnaires to owners regarding behavior and function of dogs with low back pain due to DLSS, as used in comparable studies [27,39,40], were supplied before treatment and at 6, 12, and 24 weeks (Table 1). If owners perceived lameness and disability in daily activities that was of similar (or worse) severity compared to pre-treatment level and lasted for at least 1–2 days, they were allowed to administer pain medication that the dog had received prior to inclusion. The owner contacted the study coordinators (A.R.T., T.W.) for guidance by phone or email, and if necessary, could book an extra outpatient visit. The owners were asked to register if rescue analgesia was given.

### 2.8. Statistical Analysis

Statistical analysis was performed using R (R version 4.0.4., and RStudio 1.4.1106, Vienna, Austria). Baseline characteristics were expressed as n and % for count variables, as median and IQR or range for ordinal variables, and as mean ± standard deviation (SD) for numeric variables with normal distribution. Statistical tests to describe baseline differences were not reported according to Consolidated Standards of Reporting Trials (CONSORT) guidelines [42] and statistical analysis was conducted only for the period of blinded measurements, i.e., up to 12 weeks follow-up after intradiscal injection. Normality was assumed when histograms showed symmetry and the Shapiro–Wilk test was not significant. Questionnaire data of the open-label 12 to 24 week follow-up is provided in a descriptive manner.

#### 2.8.1. Primary Outcome Measurements

Primary outcome measurement was the change in the sumscore of the improved questionnaire. The 10 item questionnaire (Table 1) was validated and improved in the current research population by (1) confirming unidimensionality using principal component analysis (PCA); (2) determining irrelevant or redundant items; and (3) item reduction when appropriate. In the PCA, sample adequacy was assumed when (1) the Kaiser–Meyer–Olkin measure of sampling adequacy was >0.7; (2) the Bartlett test yielded a significant result (*p* < 0.05); and (3) the determinant was >0.00001. Unidimensionality was assumed when (1) only one principal component was to the left of the elbow in the scree plot; (2) only one principal component was detected in parallel analysis; and (3) all items were loaded on this extracted factor with factor loading >0.35. Items were considered irrelevant when the item–rest correlation was <0.35. An item was considered redundant when it had a Pearson’s correlation coefficient >0.8 with another item. Further item reduction was achieved by elimination of items that did not cause a decrease in reliability, as expressed by Cronbach’s alpha, or a decrease in the mean inter-item correlation if this item was dropped.

As such, the primary outcome measure was based on 6 questions (Table 1), each with a 10 point scale at 6 and 12 weeks: 1. Pain or lameness of the pelvic limbs; 2. Weakness of the pelvic limbs; 3. Low back pain; 4. Difficulty rising; 5. Difficulty lying down; 8. Tail wagging. The change in questionnaire sumscore (∆questionnaire sumscore) was employed for statistical analysis.

Differences in ∆questionnaire sumscore between treatment and placebo groups were tested using the Wilcoxon rank sum test with Benjamini–Hochberg correction for multiple testing. Treatment score was defined as a minimum improvement of 10 points in sumscore, because the standard deviation of the baseline sumscore was 9.64. This minimum improvement of 10 points over a scale of 60 points corresponds with an improvement of >16% of the sumscore over time. In analogy to the Canine Brief Pain Inventory (CBPI), this change is considered clinically relevant. The CBPI questionnaire, which registers an increase of 1 point in the “severity of pain” question and 2 points in the “pain interference with daily activity” questions, corresponding to 10% and 20%, respectively, considers these changes to represent an improvement over time [43]. The differences were expressed as risk ratios for treatment success, as defined by a minimum improvement of one standard deviation from the baseline sumscore of the 6 item questionnaire at 6 and 12 weeks. Differences in risks were statistically tested using Fisher’s exact test with Benjamini–Hochberg correction for multiple testing.

Although confounding is limited in an RCT, it can be introduced by baseline imbalances in studies with a small sample size. A variable was considered a confounder when (1) it showed imbalance at baseline; (2) it had an association with the outcome; and (3) there was a biologically plausible causal pathway from this variable to the outcome. Based on this, a priori subgroup analyses on ∆questionnaire sumscore were planned for sex (male, female), disc protrusion (<25%, ≥25% into the spinal canal), and Modic changes (present, absent) in MRI at baseline.

#### 2.8.2. Secondary Outcome Measurements

Force plate analysis was performed at 6 and 12 weeks as the change from the baseline for P/T Fz+, P/T Fy+, and P/T Fy− [39]. Differences between the treatment arms in change scores were examined using the two-sample *t*-test without Welch’s correction. Equality of variances between subgroups was assumed when Levene’s test yielded no significant result. Significance was set at *p* < 0.05 for all analyses.

To examine possible adverse or beneficial effects of CXB-loaded PEAM treatment, the aforementioned descriptive categorical MRI findings commonly reported in DLSS [28] were compared with the baseline. Furthermore, at baseline, the T2 relaxation times were correlated to the Pfirrmann grade, tested for normality (Shapiro–Wilk test), and compared between CXB-PEAM and placebo groups at baseline via the non-parametric Mann–Whitney U test. Differences in time within the CXB-PEAM group were tested via the non-parametric Wilcoxon sign rank sum test. Furthermore, to assess the change in T2 relaxation time over the 3 months follow-up in relation to baseline Pfirrmann grades of the PEAM-CXB treated discs, Spearman’s rank correlation was employed and differences were analyzed via the Wilcoxon sign rank sum test.

## 3. Results

### 3.1. Baseline Characteristics

The population of dogs included in the study consisted of client-owned dogs suffering from mild to moderate DLSS, but excluding dogs with end-stage DLSS, and were second or third referral patients. Baseline characteristics are summarized in Table 2. Thirty dogs were included in the study (Appendix A), of which 20 were randomly allocated to the CXB-PEAM group and 10 to the placebo group. The mean age of the study population was slightly younger (4.8 years) than that of the literature (5.8 years) [28]. Baseline imbalances were noted for sex (50% male dogs (10 of 20) in the CXB group vs. 80% (8 of 10) in the placebo group), purpose of the dog (7/20 or 35% sports/service dog in the CXB group vs. 6/10 or 60% in the placebo group), questionnaire sumscore (median (range): 33.5 (23−56) in CXB vs. 47.5 (28–55) in placebo), P/T Fz+ (for CXB and placebo 0.61 ± 0.2 and 0.57 ± 0.1, respectively), P/T Fy+ (for CXB and placebo 0.55 ± 0.1 and 0.50 ± 0.1, respectively), and disc protrusion <25% (70% (14 out of 20) in the CXB group vs. 50% (5 out of 10) in the placebo. None of the baseline characteristics were significantly associated with the ∆questionnaire sumscore at 6 or 12 weeks. Therefore, it was considered that these variables did not confound the association between treatment and primary outcome.

### 3.2. Owner Assessment of Pain

The validation of the 10 item questionnaire in the study population showed that the questionnaire was measuring a unidimensional construct, because only one principal component could be extracted. The reliability in this population was 0.84 (as expressed by Cronbach’s alpha, 95% C.I. 0.79–0.89). Questions 6 (muscle atrophy), 7 (tail position), 9 (urinary and/or fecal incontinence), and 10 (hyperesthesia of the back), appeared to be redundant, and were dropped (Table 1). This resulted in an improved 6 item questionnaire with a similar Cronbach’s alpha 0.84 (95% C.I. 0.79–0.89) and increased mean inter-item correlation (0.47 vs. 0.35). The effect of treatment is summarized in Table 3. The changes from baseline in the 6 item questionnaire sumscore of weeks 6 and 12 (i.e., ∆questionnaire sumscore) were reported as median and range (minimum-maximum observed score). At both time points, dogs in the CXB group were significantly improved compared to dogs in the placebo group (median +9 vs. +3 at week 6, and +12 vs. −1 at week 12. *p* = 0.0302 and *p* = 0.0640, respectively). The risk ratio for treatment success was 1.90 (95% C.I. 1.24–2.91, *p* = 0.0228) at week 6 and 1.95 (95% C.I. 1.10–3.45, *p* = 0.0362) at week 12. Prior to inclusion in the RCT, all other analgesia was discontinued. Dependency on additional pain medication was registered at 6 and 12 weeks follow-up and did not differ between groups (Table 3). Dogs dependent on additional pain medication in the 0–6 week and the 0–12 week periods had significantly lower (*p* = 0.002 and *p* = 0.009, respectively) questionnaire sumscores (median 34 (range 24–50) and 31 (range 22–51)) vs. dogs not dependent on additional analgesia in the 0–6 week and the 0–12 week periods (median 53 (range 24–60) and 56 (range 26–60), respectively). As such, the use of additional analgesia did not influence treatment success. This confirms the validity of the questionnaire.

### 3.3. Subgroup-Analysis

Subgroup analysis was undertaken for sex, degree of disc protrusion, and Modic changes. The following baseline differences were noted: male dogs had significantly higher baseline questionnaire scores compared to females (median (range) 45.5 (27–56) and 31.5 (23–50), respectively, *p* = 0.00324). Dogs with Modic changes in MRI at baseline had significantly higher baseline questionnaire scores compared to dogs without Modic changes in MRI (median (range) 45 (23–56) and 32 (25–42), respectively, *p* = 0.00324). More dogs with Modic changes than dogs without Modic changes had Pfirrmann score 4 (2 out of 21 = 10% and 2 out of 3 = 67%, resp., *p* = 0.0347). Dogs with <25% disc protrusion were significantly younger, had higher P/T Fy+ values, and less often intervertebral foramen stenosis, compared to dogs with ≥25% disc protrusion (age 3.95 vs. 6.21 (*p* = 0.013), P/T Fy+ 0.56 vs. 0.48 (*p* = 0.037), and intervertebral foramen stenosis 21% vs. 73% (*p* = 0.0086)). The primary outcome measurements in the subgroups are presented in Table 4 and Figure 1. In the subgroup of male dogs, the changes in questionnaire sumscore at 6 and 12 weeks were significantly higher in the CXB group compared to the placebo group (6 weeks: +15 (range 4–20) vs. +4 (range −1–+7), *p* = 0.0119; 12 weeks: +15 (range 3–25) vs. +1 (range −7–+14), *p* = 0.0119). The improvement in the questionnaire sumscore at 12 weeks between CXB and placebo groups tended to be significant in the subgroup of dogs that showed no Modic changes in MRI at baseline (+12 (range −4–+25) vs. +1 (range −7–+14), *p* = 0.0505). After 12 weeks follow up, the label was opened and at 24 weeks follow up the owner questionnaire was repeated. In the subgroups of male and female dogs the change in questionnaire at 24 weeks was maintained at similar levels to that at the 12 week time point (Appendix A).

### 3.4. Kinetic Gait Analysis

Results of force plate analyses are summarized in (Table 5). No significant differences were noted in the P/T Fz+, P/T Fy+, and P/T Fy− between baseline and at 6 and 12 weeks after treatment, or between the CXB and the placebo groups.

### 3.5. Descriptive MRI Findings, Disc Height Index, and T2-Mapping

At baseline, variable findings were observed in the dogs consistent with DLSS. Some changes were observed over the 3 months follow-up in the CXB-PEAM-treated dogs. In Dog#9, mild right foraminal stenosis was observed due to fibrous disc tissue extrusion or minor hemorrhage that may have occurred during injection into the disc. In Dog#10, the spinal nerve swelling that was observed at baseline had resolved.

Disc height index was normally distributed and not different between CXB and placebo groups at baseline. DHI and Pfirrmann grading in the CXB-PEAM-treated group was not significantly different at 3 months follow up compared to baseline. Mean T2 relaxation times were not normally distributed based on QQ plots and Shapiro–Wilk tests (*p* = 0.0010). At baseline, T2 relaxation times of L6-7 and L7-S1 IVDs negatively correlated with Pfirrmann grading (r^2^ = 0.6964, *p* < 0.0001; Figure 2). Pfirrmann grade 1 and 2 IVDs had significantly higher T2 relaxation times compared to grade 3 and 4 (*p* < 0.0001). At baseline, L7-S1 T2 relaxation times were not different between CXB-PEAM and placebo, and tended to decrease over time in the CXB-PEAM-treated IVDs (*p* = 0.0656), whereas the T2 relaxation times of the adjacent L6-7 IVD remained the same. Within the CXB-PEAM-treated IVDs, the difference in T2 relaxation time between baseline and the 3 month follow-up time point was directly correlated with the baseline Pfirrmann grade (r^2^ = 0.5872; *p* = 0.0155); the change in T2 relaxation time tended to be significantly lower in Pfirrmann grade 2 compared to grade 4 IVDs (*p* = 0.075).

Within the context of the subgroup analysis, demonstrating a differential treatment response based on the questionnaire sumscore between male and female dogs included in the study, the outcomes of disc degeneration on imaging were explored (Appendix A). At baseline, the level of degeneration (defined as Pfirrmann grading, presence of Modic changes, severity of disc protrusion, and T2 relaxation times) appeared to be higher in female dogs, but this difference was not significantly different.

## 4. Discussion

The present study employed client-owned dogs suffering from low back pain due to degenerative lumbosacral stenosis, a common clinical entity in veterinary practice. DLSS entails degeneration of the structures of the lumbosacral junction, which can result in stenosis of the spinal canal and compression of the spinal nerves. Both the tissue degeneration and nerve compression can lead to signs of low back pain in dogs and mimics lower back pain in humans. Dogs present with a nerve root signature, similar to sciatic pain in humans, although lameness observed by dog owners can be interpreted inconsistently and does not automatically correspond with nerve root signature [44]. The degenerative changes of the IVD in MRI are reflected by the loss in T2-weighted signal intensity and often protrusion of the intervertebral disc [28]. In a placebo-controlled RCT in which the treating veterinarian and the dog owner were blinded to the treatment group, the present study demonstrated that dogs treated with local application of CXB-loaded PEAMs had twice the probability of successful recovery and clinical improvement by at least 10 points (over a total scale of 60 points) compared to dogs treated with placebo. There were no clinical adverse effects but progression of disc degeneration at the tissue level was shown by reduction of the T2 relaxation time of the CXB-PEAM IVD at 3 months follow-up. It remains to be determined whether the reduction in T2 relaxation times is related to natural progression of disc disease or specifically induced by the intradiscal application.

A single intradiscal injection of CXB-PEAMs resulted in clinical improvement. The dose ranged from 350 to 583 µg per disc (depending on the size of the dog) and corresponded to less than a 1/200 fraction of the human celecoxib daily oral dose (i.e., 200 mg) [45]). Upon intradiscal injection, it provided analgesia and improved mobility by at least 15% and 20% at 6 and 12 weeks follow up, respectively, compared to baseline, and this was further maintained at the open label 24-weeks follow up. This patient-centered outcome was not confirmed by objective gait analysis, showing a large variability in the study population. Force plate analysis is most commonly used to measure unilateral lameness (asymmetry) and is able to accurately detect lameness when 15–20% change in asymmetry is measured [46]. Low back pain in dogs does not primarily present as unilateral lameness but may lead to a shift in forces from the hind- to the frontlimbs. This can be measured by calculating the pelvic–thoracic ratio [39,41], as done in our study, but subtle changes might not exceed the sensitivity threshold. Furthermore, force plate gait analysis is conducted using a specific moment with the dog outside its domestic habitat and daily life activities. Activities that are hampered by DLSS, such as lying down, standing up, jumping, and conducting service work for owners with impaired vision are not tested during force plate analysis. The CBPI and force plate gait analysis were both valuable in evaluating treatment success but did not accurately correspond with each other, indicating that owners define treatment success predominantly based on behavior rather than lameness [44].

The beneficial effects of intradiscal application of CXB were more pronounced for the subpopulation of male dogs and in dogs with no Modic changes in MRI at inclusion in the study. Both subgroups began with relatively less pain and interference in daily life activities and improved more over time. The median Pfirrmann grading, frequency of Modic changes, and ≥25% disc protrusion was higher in female dogs, which are all MRI findings related to chronic disease, and might partially explain the worse baseline questionnaire sumscore in female dogs. However, T2 relaxation time, which is a sensitive method for measuring IVD degeneration [47], was not significantly different at baseline between male and female dogs. To the author’s knowledge, few studies on gender-related pain threshold differences have been undertaken in dogs. Normal female dogs have a higher tactile sensitivity [48,49], and contradictory findings have been reported on mechanical and heat thresholds in normal female dogs compared to male dogs [49]. In line with this, women also tend to be more sensitive to pain [50], although the clinical significance appears to be low [51] and dependent on the interaction of multiple factors (i.e., anatomical, genetic, social) [52]. As such, in light of the relatively small study population (n = 30 dogs) of this RCT, the causes of this gender difference in clinical representation, and efficacy of the CXB-PEAM in providing analgesia and reducing interference of pain in daily activities, remain to be determined. Overall, IVD degeneration-related back pain is complex and the response to local treatment may relate to the duration of the clinical signs, the gender-dependent differences in pain sensitivity, or the underlying disc degeneration process, and even with the severity of disc degeneration. It is tempting to hypothesize that dogs with more severe degenerative changes, including the presence of Modic changes, respond less favorably compared to dogs with mild IVD degeneration, as shown in human patients who have a worse prognosis when Modic changes are present [53].

No major side effects were observed for the 12 week follow-up period after intradiscal injection of CXB-loaded microspheres. To complement this finding, quantitative MRI was performed, and demonstrated a significant decreased water and proteoglycan content at 12 weeks follow-up consistent with progression of IVD degeneration [47]. In line with reports in humans, T2 relaxation times at the baseline of the study negatively correlated with Pfirrmann grading [47]. Unfortunately, only two follow up MRIs were available in the placebo group. Therefore, it remains to be determined whether the loss of water and proteoglycan content based on decreasing T2 relaxation times relates to natural progression of disc degeneration or is caused by the intradiscal injection. Discography is questioned as a clinical diagnostic procedure to demonstrate disc-related low back pain because it can induce further disc degeneration, and even more so in the relatively less degenerated discs [54]. This phenomenon relates to large contrast agent volumes causing overpressurization of the disc [55,56]. In animal models that investigate disc degeneration, relative needle size and the injected volume are essential in the induction of disc degeneration, i.e., the ratio of the needle diameter to disc height exceeding 40% [57] and injected volume exceeding at least 60% of the volume of the disc [58]. Within this context, in a pre-clinical canine IVD degeneration model, intradiscal injection with 40 μL of PEAMs using 27 G needles, corresponding to 20% disc volume and 11% disc height, was shown to be safe and did not induce additional disc degeneration [24,26]. In this cohort, the intradiscal injection involved a 27 G needle and volumes of 75–125 μL dependent on the size of dog. As such, the needle diameter was 8% (range 6–12%) relative to the disc height and the injected volume corresponded to a mean of 3% relative to the total NP volume. In the smallest IVD, the injected volume was a maximum of 10% of the total NP volume. As such, the needle size diameter and the injected volume used in this cohort appear to fit well within the safety margins deduced from the aforementioned literature [24,26,57,58]. Overall, this implies that the change in T2 relaxation times, which is inversely related to the Pfirrmann grade, probably reflects the progression of disc degeneration rather than negative effects of the CXB-PEAM injection.

The efficacy of CXB-PEAMs to also improve the proteoglycan content of the disc may depend on the severity of disc degeneration. The disc microenvironment is influenced by the severity of disc degeneration and plays a critical role in the tissue response to a therapy. This has also been acknowledged for stem cell-based therapies [59] and may explain discrepancies between the present study and preclinical work conducted with the CXB-PEAMs. In a preclinical study in which IVD degeneration was induced in Beagle dogs via partial nuclectomy 4 weeks prior to treatment, PEAMS loaded with 280 μg CXB injected per disc possibly prevented further degeneration based on increased T2 relaxation times 12 weeks after intradiscal injection. This was corroborated by an improved score in histology and restoration of the NP proteoglycan content found in a previous study [26]. Interestingly, in this study it was demonstrated that immunopositivity of NGF, a mediator of inflammation and pain, was decreased, indicating that local controlled release of CXB can exert analgesic effects. Prevention of further degeneration was not observed in the present study. Critical differences that may explain this discrepancy between the described preclinical study and the present RCT are: (a) the severity of disc degeneration at the moment of intradiscal CXB-PEAM treatment and (b) microenvironmental differences, including the cells present within the NP. The dogs included in the RCT had naturally occurring progressive disc degeneration over the course of months to years compared to the experimental Beagle dogs in the previous study, in which degeneration was induced and allowed to further develop over the course of 4 weeks. In line with this, the RCT involved IVDs with Pfirrmann grade 2–4, whereas the Beagle IVDs had Pfirrmann grade 2–3 at the moment of intradiscal application. Inherent to these differences, the NP cell populations and their biologic response may differ within the discs of Pfirrmann grade 2. These cell populations were a mix of vacuolated notochordal cells and chondrocyte-like NP cells in the dogs included in the RCT due to the fact that these were non-chondrodystrophic dog breeds, whereas Beagle IVDs contain only chondrocyte-like cells within the NP [60]. This may explain the anabolic NP response to treatment with intradiscal CXB-PEAMs observed in the young adult experimental (chondrodystrophic) Beagles compared to the lack of concrete beneficial effects at the tissue level in the non-chondrodystrophic patient population in the present study. Beneficial structural effects were observed in the group of discs with moderate to severe disc degeneration. However, limited by the low number of discs with Pfirrmann grade 4, it remains to be determined if, in later stages of degeneration, intradiscal application of CXB-PEAM has structural beneficial long-term effects in clinical application.

## 5. Conclusions

Intradiscal injection with a slow-release formation of celecoxib based on polyesteramide microspheres appeared safe and effective in reducing pain and pain interference with daily life activities in client-owned dogs, when compared to a placebo group. It remains to be determined whether intradiscal injection of a small volume of CXB-PEAMs will have the ability to halt the degenerative process in the long term or, potentially, restore the disc.

## Figures and Tables

**Figure 1 pharmaceutics-13-01178-f001:**
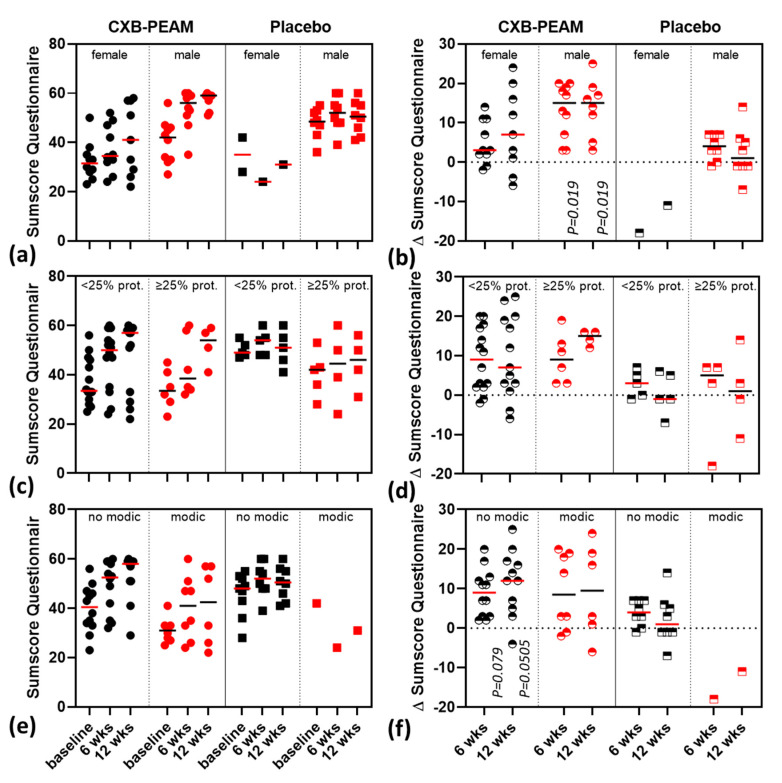
Subgroup scatterplots for questionnaire sumscore (median indicated) and corresponding change in time of the sumscore (∆sumscore) (median indicated) in 30 dogs suffering from degenerative lumbosacral stenosis treated with intradiscal injection of celecoxib-polyesteramide microspheres (CXB-PEAMs) or unloaded PEAMs (placebo). (**a**) Sumscore and (**b**) ∆sumscore for male/female demonstrating significant improvement over time within the male group of dogs with degenerative lumbosacral stenosis. (**c**) Sumscore and (**d**) ∆sumscore for degree of disc protrusion (<25% and ≥25%) were not dependent on the degree of disc protrusion. (**e**) Sumscore and (**f**) ∆sumscore Modic changes (no/yes) indicated that, within the subgroup of dogs with no Modic changes, the improvement over time tended to be significant at both time points.

**Figure 2 pharmaceutics-13-01178-f002:**
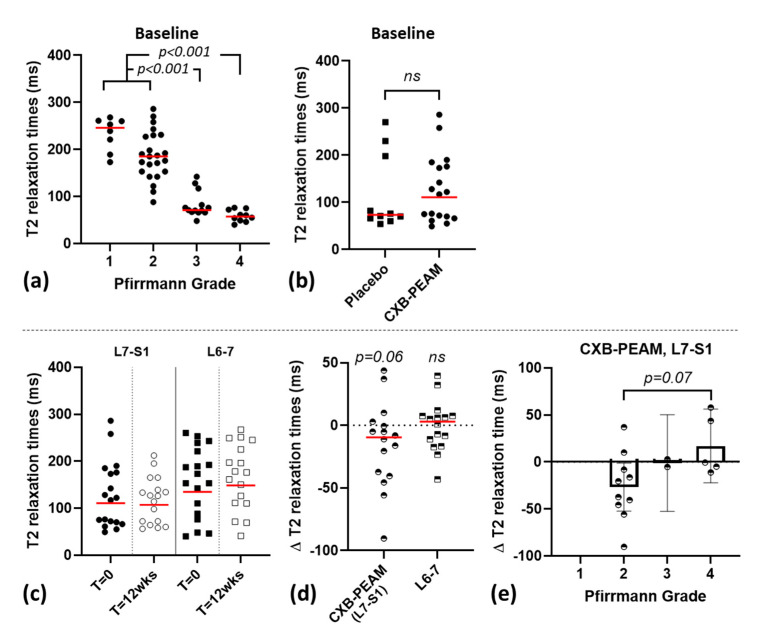
Quantitative MRI (T2 relaxation times) in 30 dogs suffering from degenerative lumbosacral stenosis, treated with intradiscal injection with celecoxib (CXB)-polyesteramide microspheres (PEAMs). (**a**) T2 relaxation time for L6-L7 and L7-S1 intervertebral discs (IVD) in relation to Pfirrmann grading at baseline. Pfirrmann grade 1 and 2 IVDs had significantly higher T2 relaxation times compared to grade 3 and 4 IVDs. (**b**) T2 relaxation time at baseline did not differ between placebo vs. CXB-PEAM-treated group. Total (**c**) and change (∆; **d**) in T2 relaxation time for L7-S1 and L6-L7 IVDs at T = 0 weeks and T = 12 weeks within the CXB-PEAM-treated group; T2 relaxation time tended to decrease over time in the CXB-PEAM-treated group. (**e**) Within the CXB-PEAM-treated discs the change (∆) in T2 relaxation time correlated to Pfirrmann grade for the L7-S1 IVD and thus hinted at improvement for grade 4 IVDs in contrast to the decrease in T2 for grade 2 IVDs (*p* = 0.07).

**Table 1 pharmaceutics-13-01178-t001:** Questionnaire to the owners of dogs with degenerative lumbosacral stenosis at baseline and 6, 12, and 24 weeks after intradiscal injection of celecoxib-polyesteramide microspheres (PEAMs) or unloaded PEAMs (placebo).

Types	Questions
Questions with a 10-point scale	1. * Does your dog have pain in the pelvic limbs and show lameness?
	2. * Does your dog show weakness in the pelvic limbs?
	3. * Does your dog have low back pain?
	4. * Does your dog have difficulty rising up?
	5. * Does your dog have difficulty lying down?
	6. How would you rate muscle volume in the pelvic limbs of your dog?
	7. How is your dog holding its tail?
	8. * Is your dog able to wag its tail?
	9. Does your dog show loss of control of urination and defecation?
	10. Does your dog show pain when you touch the lower back?

* Questions marked with an * were used for group and subgroup analysis upon validation of the questionnaire. Questions 6, 7, 9, and 10 appeared to be redundant, and were dropped. Ten point scale from worst (1) to best (10) to accommodate Dutch dog owners. Question 1 scores severity of pain; questions 2, 3, 4, 5, 8 score the interference of pain with daily activities.

**Table 2 pharmaceutics-13-01178-t002:** Baseline characteristics of 30 dogs with degenerative lumbosacral stenosis, treated with intradiscal injection of celecoxib-polyesteramide microspheres (CXB-PEAMs) (*n* = 20) or unloaded PEAMs (placebo, *n* = 10).

Baseline Characteristics		CXB-PEAMs (*n* = 20)	Unloaded PEAMs (Placebo) (*n* = 10)
Sex			
Male	*n* (%)	10 (50)	8 (80)
Female	*n* (%)	10 (50)	2 (20)
Weight (kg)	mean (sd)	27.7 (7.1)	29 (9.9)
Age (years)	median (IQR)	5.1 (2.9−6.2)	4.3 (2.2−6.2)
Dose (µL/kg)	median (IQR)	3.15 (2.9−3.3)	3.1 (2.8−3.3)
Purpose			
companion	*n* (%)	13 (65)	4 (40)
sports/service dog	*n* (%)	7 (35)	6 (60)
Baseline questionnaire sumscore	median (range)	33.5 (23−56)	47.5 (28−55)
MRI t = 0			
Pfirrmann grade			
2	*n* (%)	10 (50)	4 (40)
3	*n* (%)	3 (15)	6 (60)
4	*n* (%)	7 (35)	0 (0)
Disc protrusion			
<25%	*n* (%)	14 (70)	5 (50)
≥25%	*n* (%)	6 (30)	5 (50)
Ligamentum flavum bulging	*n* (%)	6 (30)	4 (40)
Modic changes	*n* (%)	8 (40)	1 (10)
Spinal nerve swelling	*n* (%)	4 (20)	2 (20)
Intervertebral foramen stenosis	*n* (%)	8 (40)	4 (40)
Facet joint osteoarthrosis	*n* (%)	5 (25)	2 (20)
Disc Height Index	median (IQR)	0.25 (0.24−0.28)	0.25 (0.24−0.27)
FPA t = 0			
P/T Fz+	mean (sd)	0.61 (0.06)	0.57 (0.06)
P/T Fy+	mean (sd)	0.55 (0.09)	0.5 (0.09)
P/T Fy−	mean (sd)	0.78 (0.13)	0.77 (0.23)

IQR: interquartile range; sd: standard deviation; FPA: force plate analysis; P/T: pelvic/thoracic limb ratio.

**Table 3 pharmaceutics-13-01178-t003:** Primary outcome of 30 dogs suffering from degenerative lumbosacral stenosis treated with intradiscal injection of celecoxib-polyesteramide microspheres (CXB-PEAMs) or unloaded PEAMs (placebo).

	CXB-PEAMs		Unloaded PEAMs (Placebo)	
∆questionnaire Sumscore	Median (Range)		Median (Range)	
0–6 weeks	+ 9 (−2–+20) *		+3 (−18–+7)	
0–12 weeks	+ 12 (−6–+25) ^#^		−1 (−11–+14)	
**Dependency on oral pain medication**	**N, dogs**	**%**	**N, dogs**	**%**
0–6 weeks	5	25	2	20
0–12 weeks	4	20	2	20

Change (**∆**) in questionnaire sumscore (provided as Median (range) was analyzed with the Wilcoxon rank sum test with Benjamini–Hochberg correction; * *p* < 0.05, # *p* = 0.05–0.10.

**Table 4 pharmaceutics-13-01178-t004:** Subgroup analyses of primary outcome (change in questionnaire sumscore) of 30 dogs suffering from degenerative lumbosacral stenosis treated with intradiscal injection of celecoxib-polyesteramide microspheres (CXB-PEAMs) or unloaded PEAMs (placebo).

Subgroup	CXB–PEAMs		Unloaded PEAMS (Placebo) Median (Range)	
	Median (Range)			
Sex	Male	Female	Male	Female
0–6 weeks	+15 (4–+20) *	+3 (−2–+14)	4 (−1–+7) *	−18 (−18) ^
0–12 weeks	+15 (+3–+25) *	+7(−6–+24)	1 (−7–+14) *	−11 (−11) ^
**Disc Protrusion**	**<25%**	**≥25%**	**<25%**	**≥25%**
0–6 weeks	+9 (−2–+20)	+9 (+3–+19)	+3 (−1–+7)	+5 (−18–+7)
0–12 weeks	+7 (−6–+25)	+15 (+12–+16)	−1 (−7–+6)	+1 (−11–+14)
**Modic Changes**	**no Modic**	**Modic I–III**	**no Modic**	**Modic I–III**
0–6 weeks	+9 (+2–+20) ^#^	+8.5 (−2–+20)	+4 (−1–+7) ^#^	−18 (−18) ^
0–12 weeks	+12 (−4–+25) ^#^	+9.5 (−6–+24)	+1 (−7–+14) ^#^	−11 (−11) ^

Change (**∆**) in questionnaire sumscore between CXB–PEAMs and placebo was tested with the Wilcoxon rank sum test with Benjamini–Hochberg correction. * *p* < 0.05, ^#^
*p* = 0.05–0.10, ^ *n* = 1.

**Table 5 pharmaceutics-13-01178-t005:** Force plate analyses in 30 dogs with degenerative lumbosacral stenosis, treated with intradiscal injection of celecoxib-polyesteramide microspheres (CXB-PEAMs) or unloaded PEAMs (placebo).

	CXB-PEAMs		Unloaded PEAMs (Placebo)		
	Mean	SD	Mean	SD	*p*-Value (*)
**Change 0–6 weeks**					
**∆P/T Fz+**	−0.016	0.06	0.005	0.025	0.306
**∆P/T Fy+**	−0.027	0.082	0.001	0.062	0.366
**∆P/T Fy−**	0.003	0.14	−0.049	0.19	0.418
**Change 0–12 weeks**					
**∆P/T Fz+**	−0.023	0.043	0.003	0.024	0.101
**∆P/T Fy+**	−0.017	0.084	−0.004	0.032	0.671
**∆P/T Fy−**	−0.056	0.16	−0.044	0.11	0.838

Change in pelvic/thoracic ratio (∆P/T) from baseline ratio to 6 and to 12 weeks. The change in P/T ratio for Fz+ and Fy−/+ was calculated as the P/T ratio at baseline minus P/T ratio at 6 and at 12 weeks. A decrease in P/T ratio indicates a loading shift from the pelvic to the thoracic limbs occurring in dogs that wish to unload their hind legs due to lameness or weakness. (*)Two sample *t*-test.

## Data Availability

Data will be made available on a reasonable request.

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
