# Peer review of "Prospective Evaluation of Local Sustained Release of Celecoxib in Dogs with Low Back Pain"

_pharmaceutics, 2021, doi:10.3390/pharmaceutics13081178_

Round 1
Reviewer 1 Report
This study demonstrates the intra-discal injection of celecoxib-loaded polyesteramide microspheres has beneficial effect in dogs suffering from low back pain.
The introduction section is quite long. Moreover, the discussion section is also long. It seems that there is too much background information, thereby forgetting about critically discussing their findings. Authors should shorten both sections.
I’m wondering how to diagnosis the cause of low back pain in this study. Why authors focused on the disc degeneration of L6-7 and L7-S1. There are so many asymptomatic disc degeneration according to age. Authors should explain this matter.
Reviewer 2 Report
The submitted study is a continuation of the previously published study focusing on later stages of disc degeneration treatment strategies with CXB loaded PEAMs. Previously, authors have confirmed the positive influence of the CXB loaded PEAMs in controlled canine model study with Pfirrman grade 2-3 at the time of application. Here, they focused to treat disc with Pfirrman grade 2-4 randomly. Treatment with 27G needle and volume of injection per kg body weight have been justified with relative to disc height.
The matter of the work as such is interesting. The problem of regeneration of the IVD remains unsolved, although it has been undertaken for many years by numerous groups of researchers from around the world. Nevertheless, they are still far from being entirely successful. Therefore any improvement in this direction is worth an effort. The previous observations concerning the potency of the CXB loaded PEAMs justify the further steps in this direction. Therefore, the submitted manuscript should be considered up-to-date and needed.
The experiments preformed are a logical continuation of the previous observations, and are well planned. The results are convincing, although the description in all parts of the manuscripts needs some improvement (see detailed comments below).
Detailed comments
Page 2 of 24 – line 92 -94 can be placed in discussion section.
It is recommended to reduce contents by avoiding repetitions of similar contents.
Page 3 of 24, line 145 – what is individual HPLC vials?
Figure 1: It doesn’t add any new data to the current manuscript (original research article). It can be omitted.
Justify – why characterization of CXB loaded microparticle not performed?
It is ideal to present the concentration of drug/disc/weight rather than volume of CXB loaded microparticle considered for injection.
Page 9 of 24 – line 382 –“more” word can be omitted
Elaborate the possible impact of the additional oral analgesics on T2 relaxation times even though questionnaire based inputs did not confirm the influence on treatment success.
Page 17 of 24, line 642 - 645
Limited by the low 642 number of discs with Pfirrmann grade 4, it remains to be determined if in later stages of 643 degeneration intra-discal application of CXB-PEAM has also structural beneficial effects 644 in clinical application.
Can the authors elaborate on this statement on – structural beneficial effects?
It is recommended to reduce discussion contents.
Round 2
Reviewer 1 Report
This revised manuscript answers the reviewers’ questions and addresses their concerns. It deserves attention and is worthy of publication in this journal.
Author Response
Thank you for your positive appreciation of our rebuttal and revision.
Reviewer 2 Report
Revised manuscript is presented well.
Minor comment
Page 3 of 24, line 145 – what is individual HPLC vials?
Response: separate HPLC vials were prepared for the patient study. Hereby, we could bring the PEAMs in solution just before they needed to be injected into the disc. As such, “individual vials” refers to separate vials used once for intra-discal injection. This has been further clarified in the manuscript: “Individual vials were prepared so that directly prior to intradiscal injection, PEAMs in a single vial were freshly re-suspended in 3 mL 0.9% sterile saline for intra-discal application”. (lines 110-112 of the revised manuscript). - > Please check / correct
Author Response
Page 3 of 24, line 145 – what is individual HPLC vials?
Response: separate HPLC vials were prepared for the patient study. Hereby, we could bring the PEAMs in solution just before they needed to be injected into the disc. As such, “individual vials” refers to separate vials used once for intra-discal injection. This has been further clarified in the manuscript: “Individual vials were prepared so that directly prior to intradiscal injection, PEAMs in a single vial were freshly re-suspended in 3 mL 0.9% sterile saline for intra-discal application”. (lines 110-112 of the revised manuscript). - > Please check / correct
The authors would like to thank the reviewer again for reviewing our manuscript.
We now understand the confusion with respect to this comment.
The HPLC is an abbreviation for high-performance liquid chromatography. “HPLC” is replaced by “high-performance liquid chromatography” in the manuscript (line 108-109).
The PEAMs were weighted in single vials so that prior to intradiscal injection a freshly suspension of the PEAMs for every individual patient could be made. To further clarify this “individual vials” was replaced by “single vials”. (Line 108-112)
The paragraph now reads as follows:
Once dried, the PEAMs were weighted in single high-performance liquid chromatography vials to the approximate amount of 70 mg PEAMs and γ-sterilized on dry ice [26]. CXB loading was 20% resulting in 14 mg CXB per vial. Single vials were prepared so that directly prior to intradiscal injection, PEAMs in a single vial were freshly re-suspended in 3 mL 0.9% sterile saline for intra-discal application.